# Cross-sectional Survey of Medical student Attitudes to Research and Training pathways (SMART) in the UK: study protocol

Sophie Roche , Soham Bandyopadhyay, Alexander Grassam-Rowe,
Robin Andrew Brown , Poppy Iveson, Garry Mallett , Holly Eggington,
Catherine Swales

Medical Sciences Division, University of Oxford, Oxford, UK

**Correspondence to**
Dr Sophie Roche;
sophie.l.roche@googlemail.com

## ABSTRACT

**Background** An understanding and appreciation of scientific research is a key quality of the modern clinician. Yet the Medical Schools Council has previously reported a reduction in the number of clinicians performing research. To explore the reasons for this difficulty, this multicentre, cross-sectional study aims to determine the medical student involvement and perceptions of research and research-orientated careers. It will additionally identify perceived barriers and incentives to participating in research as a student.

**Methods and analysis** This cross-sectional study of medical students at UK medical schools recognised by the General Medical Council will be administered using an online questionnaire. This will be disseminated nationally over a 2-month period through collaborative university medical school and student networks. The primary outcome is to determine the extent to which medical students are currently involved in research. Secondary outcomes include identifying the personal and demographic factors involved in incentivising and deterring medical students from becoming involved in research during medical school. This will be achieved using a selection of Likert scale, multiple-choice and free text questions. Ordinal logistic regression analysis will be performed to understand the association between specific factors and student involvement in research. This study will also characterise the proportion of medical students who are currently interested in conducting research in the future.

**Ethics and dissemination** Ethics approval has been obtained from the Medical Sciences Interdivisional Research Ethics Committee, Oxford, England. The results will be disseminated via publication in a peer-reviewed medical journal and may be presented at local, regional, national and international conferences by medical student collaborators.

## INTRODUCTION

It is of vital importance to the progress of medical innovation for clinicians to perform research as their combined academic and clinical perspectives allow the most pressing challenges in healthcare to be identified and studied. Their position affords an integrative outlook, evaluating practice objectively while advancing our collective knowledge.[1] Clinicians playing an active role in research often ensure that advances made in that research are incorporated into clinical practice as efficiently as possible.[2–4]

While it is difficult to quantify the total number of clinicians involved in research, the total number of clinical academic staff employed by UK medical schools has declined since its peak in 2010.[5] An insufficient number of individuals entering the clinical academic 'pipeline' have been identified as a contributing factor in this downward trend,[5] resulting in reduced replacement of an ageing workforce. This is surprising given that involvement in research is often seen as a desirable quality in the Curriculum Vitae (CV) of medical students and junior doctors.[6] Indeed, one survey found career success was independently associated with having conducted research as a student.[7] Furthermore, it has been shown that intercalation can have positive academic consequences.[8] Additionally, it is a General Medical Council (GMC) requirement for students to have a working knowledge of research.[9] It is therefore important to gauge the aspirations and career goals of students, as well as their

attitudes towards careers which involve research and clinical academia.

### Strengths and limitations of this study

► We will be the first to examine the attitudes of current UK medical students towards previous and future research opportunities.
► The design is limited by its snapshot nature, as we are unable to see how attitudes have or will change over time.
► It is possible that those already interested in this area will be more likely to complete the survey.

While there have been attempts to survey the barriers and enablers of progression for early career clinical academics,[10 11] there are currently no published systematic surveys of medical students which seek their views on doing research both as a student and during their career as a doctor. Furthermore, recent changes to the centralised application system have meant that research publications and additional degrees are no longer granted points as of 2023,[12 13] despite such points having been shown to be one of the primary motivating factors for medical student involvement in research.[14 15] This change has proved controversial[13 15] and several authors have raised concerns that fewer medical students will now become involved with research.

Given the need to maintain, and indeed increase, the number of clinicians involved in research, it is important to identify the factors influencing medical students in deciding whether to engage in research. Previous studies have identified intrinsic demographic factors which may affect interest in particular specialty training programmes, including academic training pathways, and research, such as economic background, ethnicity and gender.[16 17] Non-demographic influences have also been identified, including previous involvement in research and access to mentors.[16–18] Similar data are however currently lacking for medical schools in the UK, and no clear data available for the interactions between demographic and non-demographic influences. With variation in both medical school requirements for involvement in research during training and opportunities to intercalate,[19] it is important that we understand the different attitudes towards clinical academia among UK medical students. We also hope to describe the experiences that have conceivably shaped these, and the perceived barriers to pursuing research during medical school and beyond.

The Survey of Medical student Attitudes to Research and Training pathways (SMART) study is an online national questionnaire-based study. The aim of this study is to ascertain medical student perceptions of research and research-orientated careers, including perceived barriers and incentives to participating in research as a student and in following a research-orientated career. It is our hope that increasing awareness of the issues medical students face will encourage solutions to be sought by medical schools and regulators. SMART will also identify key issues for the Academy of Medical Sciences to address as part of their long-term INSPIRE strategy.[20]

## METHODS AND ANALYSIS
### Primary aim
► To determine current medical students' involvement with research.

### Secondary aims
► To characterise the group of students who are interested in research.
► To identify the factors that drive medical students to conduct research.
► To identify the perceived barriers that have historically prevented and continue to prevent current medical students from partaking in research.
► To determine if current medical students are interested in conducting research.
► To identify factors that could encourage current medical students to conduct research.

### Study design
SMART is an online, national, multicentre, questionnaire-based study focusing on medical student perceptions of research and research-orientated careers, including barriers and incentives to participating in research as a student and in following a research-orientated career. The questionnaire will be disseminated through collaborative university medical school and student networks, such as the network of INSPIRE leads across the country. It is well documented that the generic collaborative method works, and that participating students benefit from involvement.[21–26] In short, the collaborative method involves a 'snap-shot, protocol-driven, pragmatic multicentre research'[24] (p 355), approach undertaken by separate groups of trainees, or in this case students; the collaborative method also allows for greater size and power of studies.[21] The SMART study will be delivered by a team of University of Oxford medical students in clinical years 4, 5 and 6. The questions included can be found in online supplemental appendix S1. The questions were formulated by medical students at Oxford University. A brief review of the existing literature was performed to identify the gaps in knowledge and to also look at similar questionnaires and qualitative studies on the viewpoints of students and academics. This allowed an understanding of domains and items relevant to determining the aim of the project. Two students separately suggested questions which were pooled before being considered by all authors. Medical student and academic staff feedback was also sought at this point. The questionnaire has face and content validity, there is no gold standard to compare against for criterion validity. Questions were adjusted following reviewer comments to make them as non-directive as possible. Construct validity will be checked during the data collection period. Feedback will be collected from a pilot study regarding the suitability of questions, length of questionnaire and suggestions for other aspects to consider.

### Patient and public involvement
No patients or members of the public were involved in the design of the study. We will share the results with interested individuals and publicly via journal publication.

**Table 1** Project timeline

| Dates | Activity |
|---|---|
| 1 April 2021 to 30 May 2021 | Recruit collaborators at all UK universities via collaboration with INSPIRE student leads |
| 1 September 2021 to 1 October 2021 | Study pilot to be run at Oxford |
| 1 October 2021 to 30 November 2021 | Study to be modified based on pilot feedback. |
| 1 September 2021 to 1 October 2021 | Study set-up, for example, training collaborators, gaining approval for this study at each centre, providing collaborators with Qualtrics logins. |
| 1 November 2021 to 1 January 2022 | Study runs nationally for 2 months. |
| 1 January 2022 to 31 March 2022 | Analyse data and prepare manuscript. |

Extended data collection periods may be incorporated to grant flexibility to centres that may have experienced logistical obstacles to study commencement.

## Methods for recruiting participants

All medical students currently studying for a UK medical degree at a UK medical school recognised by the GMC will be eligible to participate. A list of these medical schools can be found in online supplemental appendix S2.

Medical students will be invited to participate in the study through several routes:

► Medical societies.
► Medical school mailing lists.
► Organisations focused on academic medicine, such as the Academy of Medical Sciences.
► Conferences.
► Freshers' Fairs.
► Social media.

In addition, medical students enrolled in the medical schools in online supplemental appendix S2 will be invited to collaborate in the study as regional leads as described in table 1. The maximum number of collaborators from each medical school will be 1 per year. Online supplemental appendix S3 details the participant facing information used in recruitment for the SMART study.

Collaborators will ensure that their medical school is formally engaged at an early stage of this study, and they will be primarily responsible for disseminating this questionnaire among students at their medical school. Medical school collaborators will be able to request for their own specific data and the analysis done on said data from the SMART steering committee following study completion. These data will be anonymised. Researchers will have access to these anonymous data.

## Information provided to participants

The following pieces of information will be provided to participants before taking part in the study. It will be attached to recruitment emails, and appear as the front page of the questionnaire:

► Name of the study: Cross-sectional Survey of Medical student Attitudes to Research and Training pathways (SMART) in the UK .
► Name of the principal researcher carrying out the study and information on how to contact him: Soham Bandyopadhyay, soham.bandyopadhyay@st-hildas.ox.ac.uk.
► *What is the aim of this study?* This study aims to ascertain current medical student involvement with research. We also hope to identify factors encouraging and discouraging students from partaking in research and to consider what may encourage more engagement with scientific research in the future.
► *Why have I been selected to take part?* You are being invited to take part in the questionnaire as you are a medical student currently studying for a UK medical degree at a UK medical school recognised by the GMC.
► *What do I have to do?* If you choose to participate in this voluntary survey, you will be asked to complete a questionnaire about your background, your previous exposure to research and your feelings towards a research career. This study is voluntary. If you decide not to participate this will not impact your academic standing in any way. If you decide to take part, you will be asked to complete the survey by clicking on the link below. This survey is expected to take about 10–15 min to complete, but there is no time limit and you can take as much time as you like. No background knowledge is required. We will ask for your consent for the collection and storage of data in accordance with the UK General Data Protection Regulation (GDPR) within the survey. For more information on GDPR please click on the following link: https://gdpr-infoeu/.
► *Do I have to participate?* Please note that your participation is voluntary. You may withdraw at any point during the questionnaire for any reason, before submitting your answers, by closing the browser. In cases of withdrawal from the study, no new data will be collected or linked to other data from that point on. If you do not want to answer some of the questions you do not have to, but you can still be in the study. All questions are optional. Your decision whether or not to be part of the study will not affect your academic standing or your access to university support services. If you have already submitted data and wish to withdraw from the study, please contact soham.bandyopadhyay@st-hildas.ox.ac.uk by 1 October 2021.
► *Who has approved this study?* This project has received ethics clearance through the University of Oxford's ethical approval process for research involving human participants (reference R73479/RE001).
► *How will my data be used?* Your answers will be completely anonymous, and we will take all reasonable measures to ensure that they remain confidential. Your data will be stored in a password-protected

file and may be used in academic publications. Your IP address will not be stored. If you provide us with your email address, we will delete that information at the end of the study. No answers will be linked to your email address. Research data—your anonymised answers—will be stored for a minimum of 10 years after publication or public release.

► *Who will have access to my data?* Qualtrics is the data controller with respect to the personal data they hold about you and, as such, will determine how your personal data are used. Please see their privacy notice here: https://wwwqualtricscom/privacy-statement. Qualtrics will share any email address you provide and your anonymised answers with the University of Oxford, for the purposes of research. Researchers involved in the project will have access to these anonymised data. The University of Oxford is the data controller of university email addresses, please see their privacy notice here: https://complianceadmin-oxacuk/student-privacy-policy. Responsible members of the University of Oxford and funders may be given access to data for monitoring and/or audit of the study to ensure we are complying with the guidelines, or as otherwise required by law.

► *Are there any benefits to taking part?* Despite not have any immediate individual benefits by participating in this survey, you are given the opportunity to contribute to valuable and innovative research which could be used in the future by medical universities and the world. You may find this survey an opportunity to self-reflect. There will be the option to submit email address in order to be entered into a prize draw. At the conclusion of data collection, two random participants will be awarded £50 in Amazon vouchers, two further random participants will be awarded £25 in Amazon vouchers. This will be optional, as it requires you to provide personally identifying data (ie, contact details). These will not be linked to the questionnaire answers given, and will only be used for contact regarding relevant rewards as above.

► *Will the research be published?* The findings of the study may be published in peer-reviewed journals, presented at relevant conferences and meetings and a summary of the findings will be made available on social media.

► *Are there any possible risks involved with my participation?* Some of the questions that we ask may cause upset. If you experience any distress from participating in this study, you may stop the survey at any time or skip any upsetting questions. If your distress continues after leaving the survey, we have provided a list of supportive services nationwide that can be helpful and that you might consider contacting (online supplemental appendix S4, to be linked here, and appear again at the close of the survey).

► *Who do I contact if I have a concern about the study or I wish to complain?* If you have a concern about any aspect of this project, please speak to the researcher (SB) at soham.bandyopadhyay@st-hildas.ox.ac.uk who will do his best to answer your query. The researchers should acknowledge your concern within 10 working days and give you an indication of how they intend to deal with it. If you remain unhappy or wish to make a formal complaint, please contact the Chair of the Medical Sciences Interdivisional Research Ethics Committee. Email: ethics@medsci.ox.ac.uk. Address: Research Services, University of Oxford, Wellington Square, Oxford OX1 2JD OR. The Chair will seek to resolve the matter in a reasonably expeditious manner.

► *How do I find out what was learnt in this study?* This study is expected to be completed by approximately March 2022. If you would like a brief summary of the results, please write to us by email to request information.

► *Who to contact for further details?* For any further questions or more information on the study, please contact us on the following email address: soham.bandyopadhyay@st-hildas.ox.ac.uk. Alternatively, you could contact the principal investigator (CS) at catherine.swales@ndorms.ox.ac.uk.

A representation of how this information will be presented to medical students can be found in online supplemental appendix S5.

## Financial and other rewards to participants
A prize draw involving participants who have opted to provide their contact details will take place at the conclusion of data collection. Two random participants will be awarded £50 in Amazon vouchers. Two further random participants will be awarded £25 in Amazon vouchers.

## Data collection
The online questionnaire consists of 23 quantitative and qualitative questions that use a combination of the Likert scale, multiple-choice options and free text in order to broaden the capture of sentiment nuance and improve precision in the data. The questionnaire has been sent to medical students who were not involved in creating the data collection proforma. Their responses were evaluated for potential problems, and the questionnaire was updated to ensure the questions were relevant, comprehensive and accessible. A pilot study will be performed at Oxford Medical School before this study is launched nationally. There are 952 students enrolled at Oxford Medical School. Data collection will take place at the beginning of September 2021, aiming to be complete by November 2021. Based on the experiences from the pilot, the questionnaire will be modified to improve clarity, objectivity and acceptability (online supplemental appendix S6).

## Primary outcome
To describe the extent of current medical students' involvement with research.

## Secondary outcomes
To identify and understand the reasons driving and excluding medical students from research. To identify

factors that might encourage more current students to conduct research.

## Data management

All the data will be anonymised and stored in Qualtrics. Qualtrics is a cloud-based platform, with the ability to create and customise databases. Qualtrics guarantees the highest levels of security for stored data and is compliant with regulations including GDPR and ICH E6 Good Clinical Practice.[27] The research team will have individual accounts, with access to data determined by the study administrator. Audit logs are also available to provide an overview of data access and modifications.

## Statistical analysis

We will use descriptive statistics to achieve our primary outcome in describing the extent of current medical students' involvement with research.

In order to achieve our secondary outcomes we will consider the following research questions using machine learning techniques to predict in our cohort:

Q1: What factors are associated with whether an individual in our cohort intends to pursue an academic career?

Q2: What factors are associated with a respondent's interest in undertaking (more) research in the future?

### Statistical procedure

Unless explicitly stated otherwise, assume alpha=0.05, and CI=95%. Data will be cleaned of any conflicting, impossible or corrupted data by deletion of all entries of the same individual.

### Data processing for descriptive statistics

This section seeks to provide an easy to interact with description of the responses to our survey, and the demographics of those who responded.

1. Percentages for responses by each question.
2. Median and deciles of spread for answers to appropriate question(s).
3. Mode for answers to appropriate question(s).
4. Number of missing values per question. Data will be assumed to be missing completely at random.
5. Number of individuals with one or more missing values.
6. Identify sampling weights for demographic populations using data from local sources, and national data collected about medical student demographics at each school.
7. Key qualitative themes identified with free writing texts of questions 15, 16, 17, 23. We will look to identify common sentiments, having trained multiple different team members to independently look across the data and identify recurrent themes and sentiments. Each response will be evaluated by at least two of the trial team.

### Establishing correlation across responses

The tests described in this section aim to find any predictive associations and identify likely explanatory variables.

As a method, it also allows capture of the entirety of the cohort data.

We will perform multiple correspondence analysis using sampling weights from 1.5. We will account for ordinality in data with orthogonal polynomials,[28] where appropriate, to produce a simple visualisation of correlation across variables. All interval/ratio variables will be appropriately binned to produce ordinal variables for the analysis above. If there are a low number of responses then answers to question 4 may be binned into broader ethnicities and backgrounds in order to maintain statistical power.

### Testing Q1: what factors are associated with whether an individual in our cohort intends to pursue an academic career?

We will run an ordinal logistic regression using the cumulative logit link function. We will use the ordinal responses to question 20. 'How much do you agree with the statement: "I wish to pursue an academic career or an academic training pathway".' (Likert scale of 'Disagree' through to 'Strongly agree') as our response variable, and use as predictor variables the answers to other questions. Detail on proposed model construction can be found in online supplemental appendix S7.

### Testing Q2: what factors are associated with a respondent's interest in undertaking (more) research in the future?

The methods will be the same as above using the ordinal responses to question 22. 'How strongly do you agree with the statement: "I would be interested in undertaking (more) research in the future".' (Likert scale of 'Disagree' through to 'Strongly agree') as our response variable, with answers to all questions as predictor variables as detailed in online supplemental appendix S8 model construction.

### Subgroup analysis

With a priori assumptions about significant contributors to variance, one may attempt to control for these through subgroup analysis, in categorisation of the data points by a given variable or variables.

However, to ensure validity of causal inference, one must ensure that the appropriate covariates are selected for any regression model attempting to interrogate such data.[29] We believe a priori that we may see a large degree of variance between those who have already completed research and those who have not. To account for this, we propose two models (online supplemental appendix S9, equations 3 and 4) to account for this difference and ascertain key contributors in these different groups, with the goal of more policy-relevant inferences being drawn for each subgroup.

Additional subgroup analysis may be carried out if key contributors to variance are identified in the multiple correspondence analysis, or otherwise. Additional exploratory analysis may be carried out as deemed necessary, but it will be reported as such, and will be carried out with good statistical reporting in mind.

## Authorship

In accordance with the National Research Collaborative authorship guidelines,[24] all publication outputs from SMART will be listed under a unified corporate authorship: 'SMART Collaborative'. Certain publications will include named authors on the byline as well as the group name. This will follow the example set by commendable collaboratives including STARSurg and InciSioN UK.[25 26] Anyone who has demonstrated satisfactory completion of the minimum requirements for authorship will be eligible for PubMed-citable collaborative authorship in accordance with the roles defined below.

### Writing group

Responsible for the overall scientific content, data analysis and preparation of research manuscripts.

### Steering committee

Responsible for the protocol design, project coordination and data handling.

### Collaborators

A network of medical students across all medical schools. They are responsible for leading the study regionally.

## ETHICS AND DISSEMINATION
### Ethics

Ethics approval has been obtained from Medical Sciences Interdivisional Research Ethics Committee, Oxford, England (reference R73479/RE001).

### Dissemination

The protocol will be disseminated primarily through recruited medical student collaborators. Should UK medical schools wish to see the protocol, collaborators may pass it along as well. Any publications of the protocol will be advertised through social media.

Following study completion, teleconferences will be held with all collaborators to share and discuss the data analysis undertaken and the study results. Following this, the results will be presented at local, regional, national and international conferences by medical student collaborators. A standard PowerPoint presentation and poster will be created for this purpose. All presentations will be coordinated by the SMART steering committee to avoid duplications and to ensure all conference regulations are fulfilled. In addition, the results will be disseminated via publication in a peer-reviewed medical journal. All collaborators will be given PubMed-citable collaborative coauthorship under the institutional name 'SMART Collaborative'. We will have a hybrid authorship list of named authors and the institutional collaborative.

Following publication, the manuscript can be shared by collaborators with their medical schools to feedback the study results, and to highlight the scope for expanded integration of research within medical school curricula. Medical schools can request for their own specific data and the analysis done on said data from the steering committee following study completion. The fully anonymised data set will be made publicly available.

**Contributors** The study concept and design was conceived by SB and SR. Abstract was written by GM, PI and SR. Strengths and limitations were written by SR. Introduction was written by SR, RAB and HE. Methods were written by SR. Statistical analysis was written by SR and AG-R. AG-R provided the statistical planning and analysis, and contributed to survey design. Authorship was written by SR. Ethics and dissemination was written by SR. CS and SB had a supervisory role at all stages of manuscript preparation, and helped prepare and write funding applications. All authors critically reviewed the manuscript.

**Funding** This work was supported by an INSPIRE grant from the Wellcome Trust.

**Competing interests** None declared.

**Patient consent for publication** Not required.

**Provenance and peer review** Not commissioned; externally peer reviewed.

**ORCID iDs**
Sophie Roche http://orcid.org/0000-0001-6751-0636
Robin Andrew Brown http://orcid.org/0000-0003-2999-884X
Garry Mallett http://orcid.org/0000-0001-5475-9061

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
