## [Reviewer comments · BMJ Open]

ARTICLE DETAILS

TITLE (PROVISIONAL)	A Cross-Sectional Survey of Medical student Attitudes to Research and Training pathways (SMART) in the UK; Study Protocol
AUTHORS	Roche, Sophie; Bandyopadhyay, Soham; Grassam-Rowe, Alexander; Brown, Robin; Iveson, Poppy; Mallett, Garry; Eggington, Holly; Swales, Catherine

VERSION 1 – REVIEW

REVIEWER	Gartmeier, Martin Technische Universitat Munchen, Medical Education Center
REVIEW RETURNED	18-Mar-2021

GENERAL COMMENTS	I find this study protocol clear and well written. I think it contains all necessary information and makes clear the means and goals of the planned study. I only have some minor comments: - In the questionnaire instrument, Q3 starts with "If you answered yes to Question 6" - but I think you mean Question 2 here, not Question 6.- In Q16, respondents are asked to describe the research they have already been involved in. In case they have been involved with several projects, different answers may apply - and respondents may be puzzled about how to respond. I recommend to provide further advice for this situation.
--

REVIEWER	van Braak, Marije Erasmus MC, General Practice
REVIEW RETURNED	31-Mar-2021

GENERAL COMMENTS	This study aims to describe the perspectives of medical students towards participating in research activities during their training and beyond. Given the very convincing description of the need for clinical academics, this aim has the potential to contribute to finding solutions to a rather urgent problem in medical education. I do have some suggestions, however, to improve and 'streamline' the study - mostly related to the alignment between the research aim, the questions asked, and the statistical methods used. TITLE - The title talks about student 'attitudes'. In the rest of the text, you use several other terms to refer to these: perspectives, opinions, views, etc. I would suggest to choose one term consistently. Also, I
---

wonder whether this is exactly what you address in the questionnaire - but more about that below.

ABSTRACT

- The title of the study protocol suggests that you are going to investigate attitudes towards research and training, but the training part of that is not really explicit in the abstract nor in the rest of the text. Do you need the 'training' in the title? Probably something like 'doing research during training and beyond' would clarify what you are looking at?

- P4, line 28, 'ascertain' - This sounds as if you have preliminary expectations that you would like to test. However, none of these are explicitly formulated in the rest of the text. I suggest using a different word instead of 'ascertain', e.g. describe / investigate.

- From the description of the outcome measures in the abstract I wonder what really is your aim: is it to describe the views of medical students on the incentives, barriers and likelihood for engaging in research activities in the future, or is it also to describe the degree to which they have been engaged already in such activities? It seems like you try to not only get insight into their perspective on such activities, but also in their histories of doing research or their current engagement. I understand you aim to relate these to each other, but still wonder whether your objectives cover both foci. This also applies to places in the rest of the manuscript where you talk about your aim. Maybe it could help to revise these by making them uniform (sometimes, you talk about prediction as an aim - that is beyond description, I would say, and I don't think survey data really allow for determining predictors for certain outcomes.

- There is some repetition in sections 'Outcome measures' and 'Results'. Please remove the repeated information or reword it into a few summarizing sentences in the 'Results'.

STRENGTHS AND LIMITATIONS

- p5, line 9: change into 'an opportunity' or 'opportunities'

- The second strength formulated here is not particularly relevant to this study, it could also be any other kind of multi-centre study for which this applies. I would leave this one out.

- You mention that you do not investigate the change of opinions over time. What would such longitudinal analysis yield? Is this really a limitation (does it limit you conclusions about your topic) or just a different focus for research?

- It is unclear to me whether the situation described in the fourth bullet here is a strength or limitation. Are you going to address this gap, so do you see this as a strength of the study? But: can you discover whether this relation is correlative vs causative, using the statistical methods you've proposed?

INTRODUCTION

- The rationale for describing the students' views is clear and convincing.

- p5, line 33: Are the perspectives and innovations mentioned in the sentence prior indeed an example of (can they be summarized as) leadership?

- p5, line 60: Do you expect interactions between demographic and non-demographic influences? On what basis? Would these be relevant to your research question, or not necessarily needed?

- p6, lines 3-5: I had to reread the first part of this sentence several times to understand what you mean with it. I think it may be

grammatically correct, but could you consider shortening or clarifying it a bit?

- p6, line 7-8: You talk about perceived barriers here. Essentially, whatever you elicit about incentives and barriers in the survey is 'perceived'. I would suggest to describe the aim and expected results in these terms, too. There seems to be some inconsistency in the way you talk about these 'opinions' in this respect. Even in the same sentence, for example, you talk about "the experiences that have shaped these" - and I would add: perceivedly. Whatever you elicit from your participants in this survey is their perception.
- p6, line 20: What does INSPIRE refer to?

METHODS AND ANALYSIS

- primary aim: see my earlier point about 'ascertain'
- primary aim: Do you want to elicit perceptions about MEDICAL RESEARCH, or about OPPORTUNITIES FOR DOING RESEARCH ACTIVITIES / MEDICAL RESEARCH CAREER OUTLOOKS/PATHWAYS / etc.? I would suggest to be more specific in what kind of perceptions you actually elicit (throughout the study protocol).
- The secondary aims seem much more diverse and not necessarily logically related to the primary aim. E.g. 'to characterise the proportion of medical students who would want to be involved in research in their future career', and 'to identify the scope for expanded integration of research within medical school curricula'. This is a main point that I keep coming back to (sorry for that), which I think needs some further consideration: the alignment of your primary objective and the research aims - and, furthermore, also with the questions asked in the survey and the statistical tests proposed. If your aim is to describe the perceptions of medical students about opportunities/possibilities and interests for doing research-related activities during their medical training and after that training, then determining the proportion of medical students who would indeed want to pursue a research-related career is less helpful to that aim. Also, if your aim is to describe the perceptions, statistical tests that try to predict in my view do not contribute much to that description (see later).
- p6, line 60: The study is not disseminated, the questions / the survey are.
- p7, line 5: Could you please elaborate a bit more on what you mean with 'collaborative method'?
- p7, patient and public involvement: Journal publications usually are not easily accessible to the public. Do you plan to disseminate your results to the public also in a more low-key manner?
- p7: One more general question that pops up: you describe very clearly how you want to reach out to potential survey respondents, but information on how many respondents you aim for is lacking. This is important in the context of power calculations for your statistical tests. Do you have any idea about the minimum number of respondents required?
- p8, Table 1: how do you aim to collect the feedback on the pilot survey?
- p8, Table 1: one month for getting the approval of centres to participate in your study may be too short. Have you account for any delays in this process?
- p8, Table 1: Your study will run for two months. Why did you choose that time frame?
- p8, Table 1: Two months for data analysis and manuscript preparation seems a bit unrealistic to me, based on my own

experience. Do you have time to expand this period, or do you have reasons to believe that this is indeed possible?

- p8, information provided to participants: Please adapt this information according to potential changes in the preceding text (e.g. the aim).
- p8, line 49: Each survey > is there more than one survey?
- p9, line 3-5: What will happen to the already collected data when participants withdraw from the study?
- p9, line 31-34: And also the researchers themselves will have access to the data - I think everyone will assume that, but you could add that to be complete.
- p9, last question: Personally, this sounds a bit 'heavy' to me (for example, the mention of the suicide helpline), but I do applaud your efforts to provide suggestions in case any help is needed. Great!
- p10, line 3: There is an extra space before 'Who do I...'
- p10, Data collection: What do you mean with "broaden the capture and improve granularity of the data"? Could you please motivate how the combination of the different question types contribute to that?
- p10, primary outcome: The outcome here is again slightly different from earlier versions of it in the preceding text. Do you really want to predict? If so, I doubt whether survey data are the best data collection tool. I would say that 'descriptions' of influencing factors are better suited to the possibilities of survey data.
- p11, Data management: How do users receive the individual accounts?
- p12: 'What do we hope to ascertain?' What you describe after this, are the research questions. These cannot be ascertained. This points at a major point that I have been thinking about all along: do you have any expectations regarding the predictors of pursuing an academic career? If so, what are these expectations based on? You do not describe them in the literature section, at least not in the way that it becomes clear that those are the factors that we could expect to influence medical students' choice or interest in research activities. This also relates to the questions you ask. Many of those (e.g. Q 4-10) can be read as suggestions for factors that could be at play in someone's interest in doing research. What are they based on? I would strongly encourage you to add some information about the construction of the questionnaire somewhere in the study protocol.
- p12, line 11-12: 1.5 > 1.7
- p12, lines 41-43: What would you like to do exploratory analyses about, do you have any preliminary ideas about that? And also: what do you mean with "with good statistical reporting in mind"?
- p12, lines 44-49: Maybe it is because I have no experience with the type of analyses that you describe here, but I do not understand why these analyses would be helpful to answer your research questions. Why would you want to identify similarities across variables and respondents? How does that help you understand the barriers and incentives for pursuing a research career? In my view, these analyses could be left out.
- p13, Authorship: Do all authors need to fulfill at least one or all of the roles defined?

ETHICS AND DISSEMINATION

- p13: "The study protocol will also be submitted for peer-reviewed publication." > can be left out.
- p13, line 48: What does hybrid authorship mean?

	REFERENCES  - 12. Needs an additional space before 2014. - 14. Needs a dot before 2019. AUTHORS' CONTRIBUTIONS  - "provided edits and critiqued" > critically reviewed FUNDING  - Could you provide the grant number, for completion? APPENDIX  - It is unclear to me how these questions were constructed. See an earlier comment: could you elaborate on the construction of the questionnaire somewhere in the study protocol? - Q3 refers to question 6, this should be question 2. This makes me think that you have left out questions that were present in an earlier version of the questionnaire - so in relation to the prior question, how did that process go? - Q13: It may help respondents if you could provide some examples of what you consider are research activities. What some respondents consider research-related activities, others may not think they are. See for example the things mentioned at Q16. - Q15 has a double negation. Suggestion: change 'non-compulsory' into voluntary. - Q16 would be a good candidate for an open question, since the options you mention here may be leading suggestions. The same applies to Q22. - The Table in Appendix S3 is redundant after the list of phases just above it.
--	--

REVIEWER	Johnston, Peter NHS Education for Scotland, North Deanery, Pathology
REVIEW RETURNED	01-Apr-2021

GENERAL COMMENTS	General This is a timely piece of work that addresses an existential problem around the decline in prioritisation of medicine in universities and the importance of having practising doctors involved in and running relevant research programmes in the UK. It is a large study that is well intentioned. I think there are some modifications that would strengthen the project and I have tried to make suggestions as to how to do so. I would really like to see the project succeed because identifying student need is a useful way of driving university policy in a consumerist environment. Title The title does not include reference to the context of the study, namely, UK medical schools. Abstract The abstract covers the content of the paper. I do have a question about the Participants paragraph: my reading of this includes students in offshore satellites of UK Universities as they are UK degrees and are recognised by the GMC – is this what is intended? This does change the context slightly because, although the degree is the same, the environment and experience is not likely to be the same. The other comment is that in the Results section it is stated the “study will determineresearch among students studying at universities across the UK....” This seems to me to be at odds with the previous statement and at
--

	least seems to lack clarity. How do the team intend to deal with this matter? Strengths and Limitations “We will be the first to comprehensively examine...” I would question the adverb because the study is based on the use of a questionnaire the questions of which are selective. The reference to training pathways is also unclear. Training as usually understood refers to postgraduate medical education and specifically to GMC approved programmes. The study is about undergraduate attitudes to research and does not address training as such, for example, by enquiring about schemes available across the 4 nations in UK PGME relating to opportunities to pursue research. I think, as a point of accuracy, this should be changed. The last bullet point is possibly unnecessary as it is not part of the study – and the study cannot address it as a snapshot as it classifies itself. Introduction The literature quoted is in part quite elderly and it is not clear that some of the ideas expressed have current validity. It is not an area around which I have seen much literature and if my experience is real, perhaps this should be noted to emphasise the value of the work. In terms of current practice, for example, the GMC has given outcomes for undergraduates for clinical research and scholarship (https://www.gmc-uk.org/education/standards-guidance-and-curricula/standards-and-outcomes/outcomes-for-graduates/outcomes-for-graduates/outcomes-3---professional-knowledge#26-newly-qualified-doctors-must-be-able-to-apply-scientific-method-and-approaches-to-medical-research-and-integrate-these-with-a-range-of-sources-of-information-used-to-make-decisions-for-care) which is perhaps less ambitious than the study might wish. If students read the required outcomes, the need for participation in research may be less obvious. The study appears to present the stance that we need “more of the same” in terms of career modelling. Yet career choice has been widely explored with recent advances in thinking. How can the study be used to pick up the relevance of flexibility, trainee centredness, positive learning and working environment that are generic push and pull factors in how students and trainees determine how they decide on the career they seek to follow? Hence, the context of the study seems to rest in the now of now, whilst it wishes to look to the future but does not explore these issues either in the literature or the survey tool. Methods and Analysis Primary Aim See comments under “Abstract”. Secondary Aims Third bullet – it is not clear how factors can affect trainees at different stages of training when we are looking at a cohort of current undergraduates. Can this be explained or omitted, please? The aims include aspects of “background” – free school meals are mentioned as the measure of a lack of privilege in childhood. There are other measures, for example post code of parental home. In addition, the environment in which students are being educated is relevant – opportunities may be institutional as implied here but also depend at least in part on the resources available to the student for example around finance and costs. Patient and public involvement
--	--

	I wonder if any students have been involved in the study design – useful to know and I would advocate there usefully could have been. Methods for recruiting participants Using all these routes, there is an opportunity that students will be asked to participate more than once – how will multiple surveys submitted by one person be detected and eliminated? Or possibly prevented? Information provided No issues. Financial and other rewards It would be interesting to learn the reason for the approach presented. Data collection, Primary outcome, Secondary outcomes, data management No issues. Statistical analysis What do we hope to ascertain – Q1/Q2 It might be argued these aspirations are an overstatement – the study has selected a range of potential influences but is not exclusive – surveys, it is acknowledged, rarely are. It may well therefore establish predictive associations but it is likely to “determine what predicts whether an individual ... intend to pursue an academic career”. Statistical procedures, authorship, ethics, dissemination. No issues. References Ref 17 is the same as ref 7. Number 17 does not seem to be correct as regards the context of the reference. This needs to be addressed. Contributions, funding, competing interests No issues. Appendix 1 I have a number of points about the survey tool. Q1. How will the study deal with the fact that some non-graduate entry programmes go over five years and others six. Is the assumption (as per GMC) that the readiness for practice and thus the attitudes expressed are the same at the end of the programme whenever that is. Intercalation can occur at variable times in programmes. The ordering might infer it always occurs between years three and four. Q3. The range of degree offered is incomplete – is there the possibility of a “white box” for degrees like dentistry, law, MSc, PhD? Q5. This is a short list of options and needs to be more inclusive. There is no option for not answering. Q8. Suggest define what is meant by “healthcare professional”. Does this mean professions (eg medicine, nursing, biomedical science) or professionals who work in in healthcare – all jobs? Q9. Academia – does this mean “been to university”? Or College? Or both? Or what? Suggest it should be clarified. Q10. I suggest this question should be part two in both Q8 and Q9 because as it is, it will get a number that will be meaningless because we will not know to which group the number refers. Q11. Area is an odd word – geographical area? Q13. The answer to this question is like “how long is a piece of string”. What is a little bit to one person may be a lot to another. The worry is that the outcome of this question will not be meaningful.
--	--

	Q15. What if a student is not interested? Is that a barrier? Maybe asking for reasons why students do and do not do not undertake research (as sequential questions) would be more valuable? Q17. I suggest a "white box" for "other". Q22. As Q17. I have no issues with the supporting documents (Ax 3-5) which appear standard. It might be wise to check to see that all the details are appropriate to all 4 UK nations.
--	---

VERSION 1 – AUTHOR RESPONSE

Reviewer 1

- In the questionnaire instrument, Q3 starts with "If you answered yes to Question 6" - but I think you mean Question 2 here, not Question 6.

Many thanks, this has been corrected.
Supplementary file, Appendix S1, Q3

- In Q16, respondents are asked to describe the research they have already been involved in. In case they have been involved with several projects, different answers may apply - and respondents may be puzzled about how to respond. I recommend to provide further advice for this situation.

Thank you for raising this. We want to keep this a mostly closed question for ease of analysis, but will update the wording to encourage multiple answers. We have also added an option 'other' for any options not covered.

Supplementary file, Appendix S1 q16

Reviewer 2

- The title talks about student 'attitudes'. In the rest of the text, you use several other terms to refer to these: perspectives, opinions, views, etc. I would suggest to choose one term consistently. Also, I wonder whether this is exactly what you address in the questionnaire - but more about that below.

Many thanks for your comment. For clarity we have changed opinions, views and perspectives to use the term 'attitudes' consistently when relating to our questionnaire findings.

Page 3 line 2, 5

Page 4 line 12

- The title of the study protocol suggests that you are going to investigate attitudes towards research and training, but the training part of that is not really explicit in the abstract nor in the rest of the text. Do you need the 'training' in the title? Probably something like 'doing research during training and beyond' would clarify what you are looking at?

Thank you for this comment. We argue the importance of training clinical academics in the background, and questions 20, 21 and 22 aim to look at opinions on training pathways held by medical students. The main training pathway we are bearing in mind here is the entering into the Academic Foundation Programme at F1 level.

We have also changed the wording of the strength to 'we will be the first to examine the attitudes of current UK medical students towards previous and future research opportunities'.

Page 3 line 2-3

- P4, line 28, 'ascertain' - This sounds as if you have preliminary expectations that you would like to test. However, none of these are explicitly formulated in the rest of the text. I suggest using a different word instead of 'ascertain', e.g. describe / investigate.

Many thanks, we agree and this has been changed to 'determine current medical students' involvement with research'.

Page 4 line 32

- From the description of the outcome measures in the abstract I wonder what really is your aim: is it to describe the views of medical students on the incentives, barriers and likelihood for engaging in research activities in the future, or is it also to describe the degree to which they have been engaged already in such activities? It seems like you try to not only get insight into their perspective on such activities, but also in their histories of doing research or their current engagement. I understand you aim to relate these to each other, but still wonder whether your objectives cover both foci. This also applies to places in the rest of the manuscript where you talk about your aim. Maybe it could help to revise these by making them uniform (sometimes, you talk about prediction as an aim - that is beyond description, I would say, and I don't think survey data really allow for determining predictors for certain outcomes.

Thank you for this input. We have now updated the abstract to meet the journal guidelines. We have also updated our aims to more accurately capture what we hope to find in our data collection. We agree that omniscient prediction is beyond the scope of any study, and therefore only aim to predict associations within our cohort.

Page 2

Page 4 lines 30-42

- There is some repetition in sections 'Outcome measures' and 'Results'. Please remove the repeated information or reword it into a few summarizing sentences in the 'Results'.

Many thanks for this comment. The abstract has been reformulated according to the journal abstract guidelines.

Page 2

- p5, line 9: change into 'an opportunity' or 'opportunities'

Thank you, we have made this change according to your suggestion.

Page 3 line 3

- The second strength formulated here is not particularly relevant to this study, it could also be any other kind of multi-centre study for which this applies. I would leave this one out.

Many thanks for this comment, we have removed this secondary strength.

Deletion

- You mention that you do not investigate the change of opinions over time. What would such longitudinal analysis yield? Is this really a limitation (does it limit your conclusions about your topic) or just a different focus for research?

Thank you for your comment. We agree that this is beyond the remit of our current study, but going forward would perhaps hope to expand the study into a longitudinal cohort study or similar. We would argue that it does limit the conclusions we can make about the attitudes to research and training pathways as we cannot see how these have changed over time, nor how they might change in the future.

No changes made

- It is unclear to me whether the situation described in the fourth bullet here is a strength or limitation. Are you going to address this gap, so do you see this as a strength of the study? But: can you discover whether this relation is correlative vs causative, using the statistical methods you've proposed?

Thank you for this comment. We have removed this bullet point as also suggested by another reviewer.

Deletion

- p5, line 33: Are the perspectives and innovations mentioned in the sentence prior indeed an example of (can they be summarized as) leadership?

We thank you for this pertinent point. We have changed 'leadership' to 'insight'.

Page 3 line 21

- p5, line 60: Do you expect interactions between demographic and non-demographic influences? On what basis? Would these be relevant to your research question, or not necessarily needed?

Thank you for your comment. While we do not wish to predict the outcomes to the questionnaire at this point, we are keen to see whether there are interactions between for example being of BAME and having an intercalated degree, or whether different perceived barriers to research are experienced by different demographic groups. We have added further references which we have based these ideas upon.

References 16,17,18

- p6, lines 3-5: I had to reread the first part of this sentence several times to understand what you mean with it. I think it may be grammatically correct, but could you consider shortening or clarifying it a bit?

Many thanks for this comment. We have restructured and split the sentence and hope it is now more easily understood.

Page 4 lines 10-15

- p6, line 7-8: You talk about perceived barriers here. Essentially, whatever you elicit about incentives and barriers in the survey is 'perceived'. I would suggest to describe the aim and expected results in these terms, too. There seems to be some inconsistency in the way you talk about these 'opinions' in this respect. Even in the same sentence, for example, you talk about "the experiences that have shaped these" - and I would add: perceivedly. Whatever you elicit from your participants in this survey is their perception.

Thank you for your comment. We have added 'conceivable shaped'. We have also edited our aims and outcomes to consider them in terms of perception.

Page 4 line 14 and 38

- p6, line 20: What does INSPIRE refer to?

Thank you for raising this point. INSPIRE is a scheme coordinated by the Academy of Medical Sciences to engage medical, dental and veterinary undergraduates with research.

<https://acmedsci.ac.uk/grants-and-schemes/mentoring-and-other-schemes/INSPIRE>

Reference 19 added

- primary aim: see my earlier point about 'ascertain'

Many thanks, we have reworded our primary aim to 'determine current medical students' involvement with research.

Pg 4 line 32

- primary aim: Do you want to elicit perceptions about MEDICAL RESEARCH, or about OPPORTUNITIES FOR DOING RESEARCH ACTIVITIES / MEDICAL RESEARCH CAREER OUTLOOKS/PATHWAYS / etc.? I would suggest to be more specific in what kind of perceptions you actually elicit (throughout the study protocol).

Many thanks for this suggestion. We have updated our primary aim.

Page 4 line 32

- The secondary aims seem much more diverse and not necessarily logically related to the primary aim. E.g. 'to characterise the proportion of medical students who would want to be involved in research in their future career', and 'to identify the scope for expanded integration of research within medical school curricula'. This is a main point that I keep coming back to (sorry for that), which I think needs some further consideration: the alignment of your primary objective and the research aims - and, furthermore, also with the questions asked in the survey and the statistical tests proposed. If your aim is to describe the perceptions of medical students about opportunities/possibilities and interests for doing research-related activities during their medical training and after that training, then determining the proportion of medical students who would indeed want to pursue a research-related career is less helpful to that aim. Also, if your aim is to describe the perceptions, statistical tests that try to predict in my view do not contribute much to that description (see later).

We thank you for this input. We have reconsidered our primary and secondary objectives and aims. We hope that they are now clearer and more logically related. Thank you for your help and advice in streamlining these aspects of our protocol.

Page 4 lines 32-43

- p6, line 60: The study is not disseminated, the questions / the survey are.

Thank you for this comment, we have changed 'study' to 'questionnaire'.

Page 4 line 49

- p7, line 5: Could you please elaborate a bit more on what you mean with 'collaborative method'?

Many thanks for this. We have included a brief description of the collaborative method.

Page 5 lines 4-8

- p7, patient and public involvement: Journal publications usually are not easily accessible to the public. Do you plan to disseminate your results to the public also in a more low-key manner?

Many thanks for this comment. We go into detail on our dissemination plans in our abstract and under the ethics and dissemination heading.

Page 2, Page 11 lines 33-49

- p7: One more general question that pops up: you describe very clearly how you want to reach out to potential survey respondents, but information on how many respondents you aim for is lacking. This is

important in the context of power calculations for your statistical tests. Do you have any idea about the minimum number of respondents required?

We thank the reviewer for their observations about our clear methodology. We also thank them for highlighting the usage of power calculations. To avoid a priori bias in regression methodology, we have retained our option to adjust the regression method from ordinal logistic regression with a cumulative logit link function to a partial proportional odds model. The easier interpretation of the output parameters from a cumulative logit link function over a partial proportional odds model are favourable to increase engagement with a wide audience in the literature, whilst retaining statistical validity. Given the range of variables possible in the model, and a range of possible regression constructs, calculating a priori numbers for this would be highly speculative. Given the pilot nature of this initial proposed project based on circa 900 potential students from the University of Oxford, we can then use this data to help us construct valid a posteriori power assumptions before we embark on national data collection.

No change

- p8, Table 1: how do you aim to collect the feedback on the pilot survey?

Many thanks for this comment. We have updated appendix S6 with the questions we will ask at the end of the pilot study. This will be collected in the same way as all other data.

Appendix S6

- p8, Table 1: one month for getting the approval of centres to participate in your study may be too short. Have you account for any delays in this process?

Thank you for your comment. There is flexibility built into the dates, as shown to us already by the changes we have had to make to our timeline given delays due to COVID-19. We have however modelled our timeline on previous similar projects.

Page 6, dates updated to incorporate currently experienced delays due to COVID-19

- p8, Table 1: Your study will run for two months. Why did you choose that time frame?

Thank you for your comment. There is flexibility built into the dates, as shown to us by the changes we have had to make to our timeline given delays due to COVID-19. We have modelled our timeline on previous similar projects.

Page 6, dates updated to incorporate currently experienced delays due to COVID-19

- p8, Table 1: Two months for data analysis and manuscript preparation seems a bit unrealistic to me, based on my own experience. Do you have time to expand this period, or do you have reasons to believe that this is indeed possible?

Thank you for your comment. There is flexibility built into the dates, as shown to us by the changes we have had to make to our timeline given delays due to COVID-19. We have modelled our timeline on previous similar projects.

Page 6, dates updated to incorporate currently experienced delays due to COVID-19

- p8, information provided to participants: Please adapt this information according to potential changes in the preceding text (e.g. the aim).

Many thanks for highlighting this. We have updated the information accordingly.

Page 6 lines 13-19

- p8, line 49: Each survey > is there more than one survey?

Thank you for noticing this, the mistake has been corrected to 'this survey'.
Page 6 line 28

- p9, line 3-5: What will happen to the already collected data when participants withdraw from the study?

Thank you for your comment. We have added a line to say that if participants want already collected data to be withdrawn from the study, they can get in contact with sophie.l.roche@gmail.com by 1st of October 2021 for destruction of their data.
Page 7 lines 11-14

- p9, line 31-34: And also the researchers themselves will have access to the data - I think everyone will assume that, but you could add that to be complete.

Many thanks for this comment, we have added this statement in accordingly.
Page 7 line 30-31

- p9, last question: Personally, this sounds a bit 'heavy' to me (for example, the mention of the suicide helpline), but I do applaud your efforts to provide suggestions in case any help is needed. Great! Many thanks for your supportive comments regarding this information. We understand that we cannot know how or why questions may be distressing for participants and therefore hope to err on the side of caution by providing such information.

No changes

p10, line 3: There is an extra space before 'Who do I...'.
Thank you, this space has now been removed.

Page 8 line 8

- p10, Data collection: What do you mean with "broaden the capture and improve granularity of the data"? Could you please motivate how the combination of the different question types contribute to that?

We thank the reviewer for wisely asking us to expand on this. We propose that through the question-wise appropriate use of either a Likert scale, or multiple choice, or free text, we are able to broaden the capture of sentiments and opinions of respondents, whilst increasing the precision of the responses as well. This broader capture and increased precision is compared to either binary 'yes' or 'no' questions that allow for far less nuance in answers, and far less precision in the amplitude of sentiment underlying the response. We thank the reviewer for raising this, and have changed the quoted sentence to "broaden the capture of sentiment nuance and improve precision in the data."

Page 8 line 40-41

- p10, primary outcome: The outcome here is again slightly different from earlier versions of it in the preceding text. Do you really want to predict? If so, I doubt whether survey data are the best data collection tool. I would say that 'descriptions' of influencing factors are better suited to the possibilities of survey data.

Thank you for your comment. We have updated our primary outcomes according to our updated aims.
Page 9 lines 3-9

p11, Data management: How do users receive the individual accounts?

Many thanks for your comment. Each member of the research team who needs to have access to the data can have an individual account created for them by the study administrator.

Page 9 line 15

- p12: 'What do we hope to ascertain?' What you describe after this, are the research questions. These cannot be ascertained. This points at a major point that I have been thinking about all along: do you have any expectations regarding the predictors of pursuing an academic career? If so, what are these expectations based on? You do not describe them in the literature section, at least not in the way that it becomes clear that those are the factors that we could expect to influence medical students' choice or interest in research activities. This also relates to the questions you ask. Many of those (e.g. Q 4-10) can be read as suggestions for factors that could be at play in someone's interest in doing research. What are they based on? I would strongly encourage you to add some information about the construction of the questionnaire somewhere in the study protocol.

Many thanks for your comments. We have removed the subheading 'what we hope to ascertain' and simply described the research questions. The questionnaire options have been determined from an informal qualitative gathering exercise from current medical students and will be further built upon following the pilot feedback. We have also provided an 'other' option where appropriate that will open up to a free text option to capture any factors that we did not conceive of. This free text information can be inductively coded. We have now included the construction of the questionnaire in the study protocol following your suggestion.

Page 9 lines 25-30

- p12, line 11-12: $1.5 > 1.7$

Thank you for your comment. While this was initially incorrect as you noted, due to edits 1.5 is now the correct number.

Page 10 line 11

- p12, lines 41-43: What would you like to do exploratory analyses about, do you have any preliminary ideas about that? And also: what do you mean with "with good statistical reporting in mind"?

p12, lines 44-49: Maybe it is because I have no experience with the type of analyses that you describe here, but I do not understand why these analyses would be helpful to answer your research questions. Why would you want to identify similarities across variables and respondents? How does that help you understand the barriers and incentives for pursuing a research career? In my view, these analyses could be left out.

Many thanks for this comment. The exploratory analysis would be focused around the structure and distributions within the data. That is, if our collected responses have particularly non-Gaussian distributions, or extensive collinearity, as examples of distributions and structures that would disrupt valid statistical analysis without correction, then we would need to explore the extent of this. It is difficult to predict what the structure of the data will be, and would predispose the study to confirmation bias, amongst other biases, to plan extensively for the exploratory analysis of the data structure before data collection and analysis.

On your suggestion, we have expanded on our statistical plans in Appendix S7, S8 and S9. We have also removed the K cluster approach analysis, sticking with descriptive statistics and basic modelling to account for co-linearity as described above.

Appendix S7, S8, S9

- p13, Authorship: Do all authors need to fulfill at least one or all of the roles defined

Thank you for this comment. In order to be included in the 'SMART collaborative' authorship group, an individual will need to fulfil at least one of the roles.

No change

- p13: "The study protocol will also be submitted for peer-reviewed publication." > can be left out.
Many thanks for this comment, we have deleted the sentence.

Deletion

- p13, line 48: What does hybrid authorship mean?

Thank you for your comment. Hybrid authorship pertains to a mixture of individual authors plus the collaborative group e.g. 'S. Roche, SMART Collaborative'

No change

- 12. Needs an additional space before 2014.

Many thanks, this has been corrected.

Page 13 line 42

- 14. Needs a dot before 2019.

Many thanks, this has been corrected.

Page 14 line 5

- "provided edits and critiqued" > critically reviewed

Thank you, this has been changed

Page 14 line 27

- Could you provide the grant number, for completion?

Thank you for this comment. Unfortunately INSPIRE do not provide a grant number to their recipients, otherwise we would happily include it.

No change

- It is unclear to me how these questions were constructed. See an earlier comment: could you elaborate on the construction of the questionnaire somewhere in the study protocol?

Thank you for this comment. The questions were formulated by medical students at Oxford University, with two students separately suggesting questions before being considered by all authors. Feedback will be collected from the pilot study regarding the suitability of questions and suggestions for other aspects to consider.

Page 5 lines 11-14

- Q3 refers to question 6, this should be question 2. This makes me think that you have left out questions that were present in an earlier version of the questionnaire - so in relation to the prior question, how did that process go?

Many thanks for this comment, q3 has been updated accordingly. We have also added a note on the creation of the questionnaire itself.

Supplementary file, Appendix S1, Q3

Page 5 lines 11-14

- Q13: It may help respondents if you could provide some examples of what you consider are research activities. What some respondents consider research-related activities, others may not think they are. See for example the things mentioned at Q16.

Many thanks for this comment. This was considered in the planning stages, however we thought it would be valuable to see differences in perceptions of whether 'a lot' of research to one participant was regarded differently by another and if there were any correlates with this.

No change

Q15 has a double negation. Suggestion: change 'non-compulsory' into voluntary.

Thank you for this comment. We have updated the wording accordingly.

Supplementary file, Appendix S1, Q15

Q16 would be a good candidate for an open question, since the options you mention here may be leading suggestions. The same applies to Q22.

Thank you for raising this. We want to keep this a closed question for ease of analysis, but will update the wording to encourage multiple answers. We have also added an option for 'other' for any options not covered

Supplementary file, Appendix S1 q16

The Table in Appendix S3 is redundant after the list of phases just above it.

Many thanks, this table has now been deleted

Deletion

Reviewer 3

The title does not include reference to the context of the study, namely, UK medical schools.

Many thanks for your comment, the title has been changed accordingly

Page 1, Title

The abstract covers the content of the paper. I do have a question about the Participants paragraph: my reading of this includes students in offshore satellites of UK Universities as they are UK degrees and are recognised by the GMC – is this what is intended? This does change the context slightly because, although the degree if the same, the environment and experience is not likely to be the same.

Many thanks for this comment. The abstract has been reformulated according to the journal abstract guidelines.

Page 2

The other comment is that in the Results section is that it is stated the “study will determineresearch among students studying at universities across the UK....” This seems to me to be at odds with the previous statement and at least seems to lack clarity. How do the team intend to deal with this matter?

Many thanks for this comment. The abstract has been reformulated according to the journal abstract guidelines.

Page 2

“We will be the first to comprehensively examine...” I would question the adverb because the study is based on the use of a questionnaire the questions of which are selective. The reference to training pathways is also unclear. Training as usually understood refers to postgraduate medical education and specifically to GMC approved programmes. The study is about undergraduate attitudes to research and does not address training as such, for example, by enquiring about schemes available across the 4 nations in UK PGME relating to opportunities to pursue research. I think, as a point of accuracy, this should be changed.

Thank you for this comment. We agree we cannot guarantee a comprehensive understanding. We had considered the study to include training pathways with the question ‘I wish to pursue an academic career’, as well as to a lesser extent ‘I would be interested in undertaking more research in the future’ and ‘what would encourage your involvement in research in the future’. We have changed the wording to ‘We will be the first to examine the opinions of current UK medical students towards previous and future research opportunities’.

Page 3 line 2-3

The last bullet point is possibly unnecessary as it is not part of the study – and the study cannot address it as a snapshot as it classifies itself.

Many thanks for this comment, we have removed this as a limitation.

Deletion

The literature quoted is in part quite elderly and it is not clear that some of the ideas expressed have current validity. It is not an area around which I have seen much literature and if my experience is real, perhaps this should be noted to emphasise the value of the work. In terms of current practice, for example, the GMC has given outcomes for undergraduates for clinical research and scholarship (<https://www.gmc-uk.org/education/standards-guidance-and-curricula/standards-and-outcomes/outcomes-for-graduates/outcomes-for-graduates/outcomes-3---professional-knowledge#26-newly-qualified-doctors-must-be-able-to-apply-scientific-method-and-approaches-to-medical-research-and-integrate-these-with-a-range-of-sources-of-information-used-to-make-decisions-for-care>) which is perhaps less ambitious than the study might wish. If students read the required outcomes, the need for participation in research may be less obvious.

Thank you for your comment. We agree that it is an area with limited literature, however since first writing the draft there have been a few more publications based on the recent changes to the UKFPO educational performance measures changes. We have now included these and other up-to-date references found in a brief literature search in our introduction.

Page 3 lines 34-48

The study appears to present the stance that we need “more of the same” in terms of career modelling. Yet career choice has been widely explored with recent advances in thinking. How can the study be used to pick up the relevance of flexibility, trainee centredness, positive learning and working environment that are generic push and pull factors in how students and trainees determine how they decide on the career they seek to follow? Hence, the context of the study seems to rest in the now of now, whilst it wishes to look to the future but does not explore these issues either in the literature or the survey tool.

Many thanks for your comment. We agree that careers are changing, and there is a greater presence of portfolio careers. However we feel the aim of the project is to look at issues at medical school, not the opportunities after medical school. We completely agree that a way to encourage a culture shift in medical schools is to highlight these careers, and this could also potentially change students' priorities. But a survey is not the best vehicle to do this in. A qualitative study would be better able to understand these more complex phenomena of motivations depending on the context provided to students.

No change

Third bullet – it is not clear how factors can affect trainees at different stages of training when we are looking at a cohort of current undergraduates. Can this be explained or omitted, please?

Thank you for this comment. We have updated our primary and secondary aims according to other reviewer comments to more clearly show our goals.

Page 4 lines 31-42

The aims include aspects of “background” – free school meals are mentioned as the measure of a lack of privilege in childhood. There are other measures, for example post code of parental home. Many thanks for this comment. We are aware there are many ways of marking for economic privilege, which we did consider when creating the survey. However we felt that FSM eligibility would be a more comfortable question than parental postcode in terms of maintaining anonymity of responses.

No change

In addition, the environment in which students are being educated is relevant – opportunities may be institutional as implied here but also depend at least in part on the resources available to the student for example around finance and costs.

Thank you for this important factor that we had hereto overlooked. We have now added Q1a, 'which medical school do you attend', in order to include this in our analysis.

Supplementary file, Appendix S1, Q1a

I wonder if any students have been involved in the study design – useful to know and I would advocate there usefully could have been.

Many thanks for your comment. We agree the input of students in this study is invaluable. The idea, questionnaire and protocol were all conceived of and written by current 5th and 6th year medical students at Oxford Medical School (with one graduating over the course of production and now working as an F1 doctor). We also hope to receive feedback via our pilot survey to get a wider student view.

No changes made

Using all these routes, there is an opportunity that students will be asked to participate more than once – how will multiple surveys submitted by one person be detected and eliminated? Or possibly prevented?

Many thanks for highlighting this risk. A participant's IP address will be collected by the Qualtrics mechanism, but is not being stored. If there are duplicate IP addresses, those responses can be combined. Similarly, email addresses are being collected. If there are duplicate email addresses, their responses can be combined.

No change

Financial and other rewards

It would be interesting to learn the reason for the approach presented.

Thank you for raising this. Many students complain of 'survey fatigue' due to receiving a large number of surveys for research purposes. In order to encourage students to feel they are also standing to benefit from the survey we have decided to use a small portion of the grant money to reward those who have helped in its success as well as individual participants.

No change

Statistical analysis

What do we hope to ascertain – Q1/Q2

It might be argued these aspirations are an overstatement – the study has selected a range of potential influences but is not exclusive – surveys, it is acknowledged, rarely are. It may well therefore establish predictive associations but it is likely to "determine what predicts whether an individual ... intend to pursue an academic career".

We thank the reviewer for his comments and agree that it would be overtly optimistic to assume a omniscience prediction from our survey. We have made clearer our hopes to predict associations and understand that this is only valid for our cohort. We have therefore modified our question to "What is associated with a respondent's interest in undertaking (more) research in the future?" and we believe that this also mitigates any unintended connotations with the word "predict".

Page 9 lines 23-28

Ref 17 is the same as ref 7. Number 17 does not seem to be correct as regards the context of the reference. This needs to be addressed.

Our apologies for the incorrect referencing here. We did notice this shortly after submission and informed the journal who advised we wait until after receiving reviewer comments to edit. We have corrected the referencing.

References

Appendix 1

Q1. How will the study deal with the fact that some non-graduate entry programmes go over five years and others six. Is the assumption (as per GMC) that the readiness for practice and thus the attitudes expressed are the same at the end of the programme whenever that is.

Thank you for making this important point. We have changed the options to include 'year 5 of 6', 'year 5 of 5' and 'year 6 of 6'.

Supplementary file, Appendix S1 q1

Intercalation can occur at variable times in programmes. The ordering might infer it always occurs between years three and four.

Thank you for this comment. We have moved the intercalation option to the end.

Supplementary file, Appendix S1 q1

Q3. The range of degree offered is incomplete – is there the possibility of a “white box” for degrees like dentistry, law, MSc, PhD?

Many thanks for this suggestion, we have added an option for 'other'

Supplementary file, Appendix S1 q3

Q5. This is a short list of options and needs to be more inclusive. There is no option for not answering.

We thank you for this important comment. All questions are voluntary (no compulsory questions). We have also changed the input to include a free-text box to enable participants to self-identify in the most accurate way. This is an approach recommended by Stonewall

https://www.stonewall.org.uk/sites/default/files/do_ask_do_tell_guide_2016.pdf

Supplementary file, Appendix S1 q5

Q8. Suggest define what is meant by “healthcare professional”. Does this mean professions (eg medicine, nursing, biomedical science) or professionals who work in in healthcare – all jobs?

Many thanks for helping us to increase the clarity of our questions. We will include a definition of healthcare professional in the questionnaire.

Supplementary file S1, Q8

Q9. Academia – does this mean “been to university”? Or College? Or both? Or what? Suggest it should be clarified.

Thank you for helping us to increase the clarity of our questions. We have changed the wording to 'held an academic position' and will include a definition of such in the questionnaire.

Q8b, Q9b

Q10. I suggest this question should be part two in both Q8 and Q9 because as it is, it will get a number that will be meaningless because we will not know to which group the number refers.

Thank you for this suggestion. We will consider the best way to format this as suggested when inputting to our online interface.

Supplementary file, Appendix S1 q8b and 9b

Q11. Area is an odd word – geographical area?

Thank you, we have made this change.

Supplementary file, Appendix S1 q11

Q13. The answer to this question is like “how long is a piece of string”. What is a little bit to one person may be a lot to another. The worry is that the outcome of this question will not be meaningful.

Many thanks for this comment. This was considered in the planning stages, however we thought it might be valuable to see differences in perceptions of whether ‘a lot’ of research to one participant was regarded differently by another and if there were any correlates with this.

No change

Q15. What if a student is not interested? Is that a barrier? Maybe asking for reasons why students do and do not do not undertake research (as sequential questions) would be more valuable?

We thank you for this interesting comment. We would feel that not being interested in research is indeed a barrier. We then explore reasons for undertaking research in Q17.

No changes

Q17. I suggest a “white box” for “other”..

Many thanks for this suggestion, we will make this change

Supplementary file, Appendix S1 q17

Q22. As Q17.

Many thanks for this suggestion, we will make this change

Supplementary file, Appendix S1 q22

I have no issues with the supporting documents (Ax 3-5) which appear standard. It might be wise to check to see that all the details are appropriate to all 4 UK nations

Thank you - we can confirm that we have checked and amended details as appropriate for all 4 nations.

Supplementary file, Appendix S5

VERSION 2 – REVIEW

REVIEWER	van Braak, Marije Erasmus MC, General Practice
REVIEW RETURNED	14-Jun-2021
GENERAL COMMENTS	The authors have done a great job in further improving the protocol, addressing the issues raised by the reviewers and editor. I applaud them for the way they addressed the comments, this has indeed strengthened the study.

	I have only a few minor suggestions. The first is related to the methods description. In their response, the authors have considered the suggestion to add some information about the construction of the questionnaire. They have done so, but only to a limited extent. I suspect it would be very hard to repeat the approach and come to the same set of questions - as many directions are possible. Could the authors give a few more hints about how exactly the students went about devising the questions? Did they use other questionnaires or thematic lists or something else to direct that process? And what were the criteria for inclusion of the questions that the students came up with? Did that involve considerations of length of the questionnaire? And did you consider the directive nature of some questions, for example? Furthermore, a few minor points:  - p. 6 line 4: have prevented > or maybe also still prevent? - p. 6 line 42: 4,5 and 6 > 4, 5 and 6 - p. 11 and further: RQs start with "What is associated with..." > I would suggest changing all questions starting with this phrase into "What factors are associated with..." or any other noun that describes the 'things' that are associated with the outcomes you are researching. - p. 12 line 21: I wouldn't say the 'section aims' to..., the tests described in that section are aimed to...
--	--

REVIEWER	Johnston, Peter
	NHS Education for Scotland, North Deanery, Pathology
REVIEW RETURNED	15-Jun-2021

GENERAL COMMENTS	bmjopen-2021-050104.R1 comments The revision seems to have taken major concerns into consideration and has addressed most of them, for which many thanks. My wish is to strengthen this project to provide evidence that will be effective in supporting the academic aspects of medicine and so I have taken the liberty to add some additional observations which, given the timing, may well be considered as advisory. Re-reading it, I am still not quite clear about a couple of things. The aims are  • To determine current medical students' involvement with research • To identify factors that could encourage current medical students to conduct research The questionnaire is all about research save Q20: "I wish to pursue an academic career or an academic training pathway". I am content with this but the introduction sets the context of the work in and around the specific professional group of clinical academics. The issues this brings are  • A lot of medical research is carried out by NHS staff without input from university employed medical doctors, although clinical academics drive and are involved in many projects; • Clinical academics have generally at least three strands of work – research, teaching and clinical work. Is the study about research or about clinical academic medicine? The interest in research is in itself of value and there is clear need to drive ambition to carry out research in and with the NHS or other healthcare organisations internationally. The linkage is sound. How we do this and how we support organisations to do it will be influenced by pressure from staff to be able to participate in
---

	research and so there is value in this regard from the data that ill come from this study. Being a clinical academic, though, is a different question. First of all, clinical academics in the UK have different terms and conditions including (possibly significantly) pension arrangements. I would argue that the perception in the UK is they are expensive – on a clinical salary scale because retention would be on very shaky ground if it were not the situation – reflects more the poor remuneration of non-clinical academic staff in UK universities. There is, however, no clear support from UK universities to increase the number of clinical academics (maybe better to say there is clearly no support....). So, my perennial question is what career opportunities are we encouraging students to follow when the pressures to perform highly across at least three fields are greater, the Ts and Cs are worse and there is an apparent lack of employing institutional support for the job? These paragraphs begin to develop two separate arguments through which research is a common thread but which it does not unite. And so, I agree clinical academic medicine is under threat. I agree research needs to be supported and developed and that students need to be encouraged to develop their full potential across a range of possible activities (and yes the subtext is I feel we train doctors now to be doers not educate to be thinkers). The study aims focus on research; the questionnaire majors on research. And so - and I regret this – the pitch about clinical academics is not quite right in this context although I am quick to say I agree with all the sentiments expressed about that group. I guess it is hard to find out how many projects have medical PIs and how many papers are first authored by clinically active doctors but such information would maybe provide useful background. I am not sure how to weave the clinical academic issue into this to make a convincing story. And I regret not seeing this more clearly on my first review. Two small points Are ethics happy with the use of a gmail address as I note this is change – the servers are not owned by any academic organisation and security may be an issue (as in Doodle polls) and so there is a possible issue as to who owns data coming via that route. Q20 “I wish to pursue an academic career or an academic training pathway” I could suggest these are two distinctly different questions because many follow the latter but do not go on to be clinical academics. Also when we say academic training pathway, do we mean pre or post graduation and then what?
--	--

VERSION 2 – AUTHOR RESPONSE

viewer 2	Could the authors give a few more hints about how exactly the students went about devising the questions? Did they use other questionnaires or thematic lists or something else to direct that process?	Thank you for inviting us to expand on the formulation of the questionnaire. We hope that it is now clearer how we came to the final set of questions. ‘The questions were formulated by medical students at Oxford University. A	P5 lines 1-15
----------	--	---	---------------

	And what were the criteria for inclusion of the questions that the students came up with? Did that involve considerations of length of the questionnaire? And did you consider the directive nature of some questions, for example?	brief review of the existing literature was performed to identify the gaps in knowledge and to also look at similar questionnaires and qualitative studies on the viewpoints of students and academics. This allowed an understanding of domains and items relevant to determining the aim of the project. Two students separately suggested questions which were pooled before being considered by all authors. Medical student and academic staff feedback was also sought at this point. The questionnaire has face and content validity, there is no gold standard to compare against for criterion validity. Questions were adjusted following reviewer comments to make them as non-directive as possible. Construct validity will be checked during the data collection period. Feedback will be collected from a pilot study regarding the suitability of questions, length of questionnaire and suggestions for other aspects to consider.	
	- p. 6 line 4: have prevented > or maybe also still prevent?	Many thanks for this helpful suggestion. Changed to 'historically and continue to prevent current medical students from partaking in research'	Page 4, line 28
	- p. 6 line 42: 4,5 and 6 > 4, 5 and 6	Many thanks, we have added this space in.	P4 line 49
	- p. 11 and further: RQs start with "What is associated with..." > I would suggest changing all questions starting with this phrase into "What factors are associated with..." or any other noun that describes the 'things' that are associated with the outcomes you are researching.	Thank you, we agree this is better wording and have made the change as suggested.	P9 lines 20 and 22 P10 lines 12 and 20

	- p. 12 line 21: I wouldn't say the 'section aims' to..., the tests described in that section are aimed to...	Many thanks for this suggestion. We have made the change 'The tests described in this section aim to find...'	P10 line 2
Reviewer 3	Is the study about research or about clinical academic medicine? I guess it is hard to find out how many projects have medical PIs and how many papers are first authored by clinically active doctors but such information would maybe provide useful background. I am not sure how to weave the clinical academic issue into this to make a convincing story. And I regret not seeing this more clearly on my first review.	Thank you for pointing out that motivating factors and barriers to pursuing a formal clinical academic career goes beyond the purpose and aim of this paper. We have taken note of this comment and focused the introduction of the importance of conducting research among all clinicians.	Abstract – background P3 Introduction
	Are ethics happy with the use of a gmail address as I note this is change – the servers are not owned by any academic organisation and security may be an issue (as in Doodle polls) and so there is a possible issue as to who owns data coming via that route.	Thank you for highlighting this. Unfortunately as Sophie graduates this year she will no longer have access to the university email address, however we did not properly consider the important consequences of using a gmail address. We have changed these contact details to those of another researcher approved by ethics who has a longer-term university address, soham.bandyopadhyay@st-hildas.ox.ac.uk	P6 line 13 P7 line 4 P8 line 1,2, 15 Supplementary appendix S3 and S5
	Q20 “I wish to pursue an academic career or an academic training pathway” I could suggest these are two distinctly different questions because many follow the latter but do not go on to be clinical academics. Also when we say academic training pathway, do we mean pre or post graduation and then what?	Many thanks for this suggestion, we agree and have split the question. By academic training pathway we mean the post-graduate academic foundation programme, academic clinical fellow, clinical lectureship, and fellowship.	Supplementary appendix S1 Q20, Q21

--	--	--	--

VERSION 3 – REVIEW

REVIEWER	van Braak, Marije Erasmus MC, General Practice
REVIEW RETURNED	13-Jul-2021

GENERAL COMMENTS	I have no further comments, the authors have done a neat job of addressing all reviewers' comments. In my view, this study protocol is ready for publication.
---

REVIEWER	Johnston, Peter NHS Education for Scotland, North Deanery, Pathology
REVIEW RETURNED	29-Jun-2021

GENERAL COMMENTS	Many thanks. I think the revisions have dealt with any ambiguity about what the study is seeking to establish. I hope this makes the RQ clearer and again am sorry I did not spot this in the first review. Best wishes with the study and I will look out for the results in due course!
---